# Co-Benefits of Largescale Organic farming On huMan health (BLOOM): Protocol for a cluster-randomised controlled evaluation of the Andhra Pradesh Community-managed Natural Farming programme in India

Lindsay M. Jaacks[1,2]*, Lilia Bliznashka[1], Peter Craig[3], Michael Eddleston[4], Alfred Gathorne-Hardy[1], Ranjit Kumar[5], Sailesh Mohan[2], John Norrie[6], Sheril Rajan[2], Aditi Roy[2], Bharath Yandrapu, Nikhil Srinivasapura Venkateshmurthy[2,7], Poornima Prabhakaran[2]

1 Global Academy of Agriculture and Food Systems, The University of Edinburgh, Midlothian, United Kingdom, 2 Public Health Foundation of India, New Delhi, India, 3 MRC/CSO Social and Public Health Sciences Unit, University of Glasgow, Glasgow, United Kingdom, 4 Centre for Cardiovascular Science, The University of Edinburgh, Edinburgh, United Kingdom, 5 Agribusiness Management Division, ICAR-National Academy of Agricultural Research Management, Hyderabad, India, 6 Usher Institute, The University of Edinburgh, Edinburgh, United Kingdom, 7 Centre for Chronic Disease Control, New Delhi, India

* lindsay.jaacks@ed.ac.uk

**Data Availability Statement:** No datasets were generated or analysed during the current study. All

## Abstract

The BLOOM study (co-Benefits of Largescale Organic farming On huMan health) aims to determine if a government-implemented agroecology programme reduces pesticide exposure and improves dietary diversity in agricultural households. To achieve this aim, a community-based, cluster-randomised controlled evaluation of the Andhra Pradesh Community-managed Natural Farming (APCNF) programme will be conducted in 80 clusters (40 intervention and 40 control) across four districts of Andhra Pradesh state in south India. Approximately 34 households per cluster will be randomly selected for screening and enrolment into the evaluation at baseline. The two primary outcomes, measured 12 months post-baseline assessment, are urinary pesticide metabolites in a 15% random subsample of participants and dietary diversity in all participants. Both primary outcomes will be measured in (1) adult men ≥18 years old, (2) adult women ≥18 years old, and (3) children <38 months old at enrolment. Secondary outcomes measured in the same households include crop yields, household income, adult anthropometry, anaemia, glycaemia, kidney function, musculoskeletal pain, clinical symptoms, depressive symptoms, women's empowerment, and child growth and development. Analysis will be on an intention-to-treat basis with an a priori secondary analysis to estimate the per-protocol effect of APCNF on the outcomes. The BLOOM study will provide robust evidence of the impact of a large-scale, transformational government-implemented agroecology programme on pesticide exposure and dietary diversity in agricultural households. It will also provide the first evidence of the nutritional, developmental, and health co-benefits of adopting agroecology, inclusive of malnourishment as well as common chronic diseases.

relevant data from this study will be made available upon study completion.

**Funding:** LMJ received award MR/T044527/1 from the Medical Research Council/UK Research and Innovation and additional support from the Scottish Funding Council. PC acknowledges core funding from the UK Medical Research Council (MC_UU_00022/2) and the Scottish Government Chief Scientist Office (SPHSU17). The funders did not play a role in the study design and will not play a role in the collection, management, analysis, or interpretation of data; writing of the final report; or decision to submit the final report for publication.

**Competing interests:** The authors have declared that no competing interests exist.

**Trial registration: Study registration:** ISRCTN 11819073 (https://doi.org/10.1186/ISRCTN11819073). Clinical Trial Registry of India CTRI/2021/08/035434.

## Introduction

The world is not on track to meet Sustainable Development Goal 2 and end all forms of malnutrition by 2030 [1]. Despite decades of research, including hundreds of randomised controlled trials [2], and investments in nutrition-specific programmes such as micronutrient supplements [1], almost a quarter of children around the world are stunted (short length/height-for-age) [1]. India is home to 18% of the world's population [3] but 31% of the world's stunted children [4]. The most recent national data suggest that stunting is increasing in many Indian states, including several wealthy states (e.g., Goa increased from 20% to 26%, 2015–2016 to 2019–2020) and plateauing at high levels in other states including Andhra Pradesh where the prevalence is 31% [5]. At the same time, on the opposite end of the malnutrition spectrum, obesity and nutrition-related chronic diseases such as type 2 diabetes are rising, unabated, in nearly every country, including India [1]. The second-highest population-based incidence rate of type 2 diabetes in the world was reported in Chennai, a city in south India: 20.2 cases per 1,000 population [6] compared to 5.2 cases per 1,000 population in the United Kingdom, for example [7]. There is an urgent need to understand the role of and address upstream determinants of nutrition and health.

Agriculture is one such upstream determinant. Production is at the heart of food systems, determining the nutrients and chemicals consumed by populations. Moreover, agriculture is the biggest single employer of the world's population: 27% of adults globally, 59% in low-income countries, and 43% in India are employed in agriculture [8]. Agriculture has also changed substantially over the past 50 years with the introduction of Green Revolution technology including hybrid seeds, synthetic chemicals, and mechanisation and industrialisation of farming. In India, these technologies were widely disseminated and adopted from the 1960s, resulting in large-scale monocropping of paddy and wheat and the widespread use of expensive industrial products particularly chemical fertilisers and hazardous pesticides [9]. Today, India is the second-largest consumer of organophosphate pesticides in the world [10]. In 2019–2020, organophosphates accounted for 68% of tonnes of technical grade insecticides produced in India [11] and the market segment for organophosphates is predicted to continue increasing [10]. A cross-sectional survey of approximately 900 farmers in Andhra Pradesh, a state in south India, found that 91% used synthetic pesticides including 26% reporting the use of monocrotophos, a World Health Organization (WHO) Class 1b (highly hazardous) organophosphate insecticide [12].

In this context, Andhra Pradesh, a state with a population of ~49 million including ~6 million farmers [13], passed a government order known as 'Zero-Budget Natural Farming (ZBNF)' in 2015–2016 [14]–now formally, 'Andhra Pradesh Community-managed Natural Farming' (APCNF). The APCNF programme focuses on eliminating synthetic chemical inputs and improving soil health, whilst promoting locally available bio-resources, crop diversity, and the use of indigenous plant varieties [15]. The programme aims to reach all 6 million farmers in Andhra Pradesh and stay engaged with them to achieve 100% 'chemical-free agriculture' across the state by 2034.

Agriculture can impact nutrition and health through multiple pathways [16]. One pathway is through exposure to hazardous pesticides (Table 1). Pesticide exposures in agricultural

**Table 1. Summary of evidence regarding health outcomes hypothesised to be influenced by a nutrition- and gender-sensitive agroecological programme.**

| Metric | Health outcome | Hypothesised pathways | Evidence |
|---|---|---|---|
| Fasting plasma glucose | Type 2 diabetes | Reduced exposure to organophosphate insecticides<br>Increased dietary diversity | Exposure to organophosphate insecticides has been associated with impaired gluconeogenesis and insulin resistance through oxidative stress and inflammation [33, 34]. Consumption of organic food has been associated with lower risk of type 2 diabetes [35, 36]<br>Greater dietary diversity has been associated with a lower risk of type 2 diabetes in prospective studies [37], and in a large cross-sectional study in India [38] |
| Blood pressure | Hypertension | Increased dietary diversity | Greater dietary diversity has been associated with a lower risk of hypertension in cross-sectional studies [37, 39, 40], including a large cross-sectional study in India [38] |
| Estimated glomerular filtration rate<br>Urine protein-to-creatinine ratio | Chronic kidney disease | Reduced exposure to organophosphate insecticides<br>Increased dietary diversity | Exposure to organophosphate insecticides has been associated with impaired kidney function [41, 42]<br>Greater dietary diversity could indirectly reduce the risk of chronic kidney disease by reducing risk of diabetes [37] and hypertension [37, 39, 40], which are the leading risk factors for chronic kidney disease [43] |
| Body mass index | Undernutrition and obesity | Reduced exposure to organophosphate insecticides<br>Increased dietary diversity | Exposure to organophosphate insecticides has been associated with increased risk of obesity [44], potentially through inhibiting diet-induced thermogenesis in brown adipose tissue [45] or influencing the integrity of the gut barrier and gut microbiome [46]<br>The relationship between dietary diversity and undernutrition or obesity is mixed, with a systematic review finding a similar proportion of studies with favourable, mixed, or null associations for both anthropometric outcomes [47] |
| Haemoglobin | Anaemia | Increased dietary diversity<br>Increased women's empowerment | Low dietary diversity is associated with increased risk of anaemia [47, 48], including in cross-sectional studies in India [49, 50]<br>Greater women's empowerment could indirectly reduce the risk of anaemia by increasing dietary diversity [51, 52], though independent effects of women's empowerment through mechanisms other than improvements in dietary diversity have also been documented in India [52, 53] |
| Length/height-for-age z-score in children | Stunting | Reduced exposure to organophosphate insecticides<br>Increased dietary diversity<br>Increased women's empowerment | A limited number of studies have evaluated the association of pesticide exposure and child growth, and findings have been inconclusive to date [54]<br>Greater dietary diversity has been associated with reduced risk of stunting in children [55]<br>Some studies have shown that children of more empowered women have a lower risk of stunting [56–58], but the evidence is inconclusive to date due to poor study designs [59] |
| CREDI and PEDS:DM score | Early child development | Reduced exposure to organophosphate insecticides<br>Increased dietary diversity<br>Increased women's empowerment | Exposure to organophosphate insecticides has been associated with suboptimal child development [27]<br>Dietary diversity is associated with lower risk of suboptimal child development [60–66], though the association is relatively small and does not hold across all domains of development (e.g., cognitive, socioemotional, physical, and literacy-numeracy) [67]<br>Children of more empowered women are less likely to have suboptimal child development [56] |
| Self-reported musculoskeletal pain | Musculoskeletal disorders | Increased manual labour requirements | Musculoskeletal disorders are one of the most important occupational hazards in the agricultural sector [68], and are higher among organic farmers versus conventional farmers in Thailand [69] but not in Finland [70] |
| Self-reported depressive symptoms | Depression | Reduced exposure to organophosphate insecticides<br>Increased workload, mental stress from adoption of new skills, and greater financial uncertainty | A history of pesticide poisoning has been associated with increased risk of depression among farmers [22, 71, 72]<br>Studies comparing depressive symptoms between organic and conventional farmers in high-income countries have found mixed results [70, 73–75] |

Abbreviations: CREDI, Caregiver-Reported Early Development Instruments; PEDS:DM, Parents' Evaluation of Developmental Status: Developmental Milestones.

households have been linked to several adverse health effects. Whilst cancer has been the most studied outcome to date due to the establishment of several large prospective cohorts of agricultural communities such as the Agricultural Health Study in the United States, initiated in 1993, and the AGRIculture and CANcer (AGRICAN) study in France in 2005–2007, pesticide exposures have also been linked to adverse metabolic, respiratory, and neurological effects in adults [17, 18]. A meta-analysis of 22 studies found a pooled odds ratio for diabetes of 1.58 comparing the top and bottom tertile of any pesticide exposure [8]. In India, one cross-sectional study found that rural adults with detectable levels of several organophosphate pesticides were significantly more likely to have diabetes compared to those with non-detectable levels, with odds ratios (95% confidence intervals) ranging from 1.18 (1.07–1.42) for chlorpyrifos to 2.36 (1.37–4.09) for monocrotophos [19]. However, the levels of exposure reported in that study were unprecedented and have been called into question [20]. Neurological effects, particularly dizziness and fatigue [21], have also been reported for organophosphate pesticides, which are acetylcholinesterase inhibitors, and thus exposure to these chemicals impairs the breakdown of acetylcholine, an important neurotransmitter. Organophosphate pesticide exposure (assessed by acetylcholinesterase inhibition) was also associated with suicide risk in a rural village in Mexico [22]. Inhalation of pesticides is thought to cause or exacerbate respiratory symptoms in agricultural workers, including cough, nasal allergies, hay fever, breathlessness, and chest tightness [23].

In children, pesticide exposure has been linked to impaired growth and development. Cross-sectional studies of self-reported history of pesticide exposure and stunting in Indonesia and Nepal have reported strong, positive effects [adjusted odds ratios (95% confidence intervals) of 3.90 (1.15–13.26) and 3.51 (1.33–9.23), respectively [24, 25] but no effect was observed between pesticide exposure during pregnancy assessed by urinary biomarkers and stunting in children in Bangladesh [26]. There is much more evidence regarding the association between pesticides, particularly organophosphate pesticides, and neurodevelopment. The latest systematic review on the topic identified 50 studies (none in India), 45 of which found significant adverse effects of pesticide exposure on child neurodevelopment [27].

Another pathway through which agriculture can impact nutrition and health is dietary intake (Table 1). Many studies over the past 20 years have demonstrated positive impacts of nutrition-sensitive agricultural programmes on child dietary diversity and nutritional status, and a growing number of studies have demonstrated positive effects on maternal dietary diversity and nutritional status [28]. The two major mechanisms through which agriculture can impact dietary intake are (1) agricultural production, which increases food access, and (2) agricultural income, which can increase food expenditures [29]. Depending on the specific programme activities, four other mechanisms include increasing the availability and affordability of food in local markets; women's empowerment; knowledge of nutrition, health and WASH (water, sanitation and hygiene); and strengthening of local institutions [29].

Most rigorous evidence comes from homestead food production and other home gardening programmes with very few studies of agroecology programmes. Moreover, no studies have explored the impact of such programmes on pesticide exposure and resultant health effects, nor the impacts on the double burden of malnutrition or child development. Indeed, to the best of our knowledge, only one previous nutrition-sensitive agroecology programme has been evaluated [30]. The study was conducted in rural Tanzania, where Green Revolution technologies such as synthetic pesticides are not widely used, and so the intervention was compared to subsistence farming, in contrast to India where 'conventional farming' involves nearly universal use of pesticides, as mentioned above. For the study in Tanzania, researchers trained and paid 'mentor farmers' to visit intervention farmers' households and train them in agroecological practices [30]. They also provided intervention households with legume seeds. These

interventions resulted in significant improvements in child dietary diversity, women's dietary diversity, women's empowerment, and women's depression, but did not significantly improve child growth. However, the researchers did not measure the impact of the intervention on pesticide exposure, crop yields, household income, child development, or other health outcomes such as chronic diseases.

Lastly, agriculture can improve maternal and child health and nutrition through improvements in women's empowerment (Table 1) [31]. Agricultural programmes targeted to women often promote women's access to and control over agricultural resources, including income, as well as women's decision-making in agriculture. Cross-sectional evidence consistently shows that women's empowerment is associated with improved maternal and child health and nutrition outcomes [31]. Recent evidence from a nutrition- and gender-sensitive agricultural programme conducted in rural Burkina Faso demonstrated that women's empowerment mediated effects on child stunting [32]. Although the role of women's empowerment as a pathway from agriculture to health and nutrition is now fairly well understood in agricultural contexts [31], much less is known about the mediating and mitigating role of women's empowerment in the context of agroecological programmes. Apart from the Tanzania study just discussed [30], to our knowledge no other study has assessed the effect of an agroecological programme on women's empowerment and no studies have assessed women's empowerment as a pathway through which agroecological programmes achieve impact.

The BLOOM study (co-Benefits of Largescale Organic farming On huMan health) aims to determine if there are nutritional and health co-benefits to a community-based agroecology programme implemented by a state government in India. To achieve this aim, a community-based, cluster-randomised controlled evaluation of APCNF will be conducted in 80 clusters (40 intervention and 40 control) across four districts of Andhra Pradesh in south India. Approximately 34 households per cluster will be randomly selected for screening and enrolment into the evaluation at baseline. The two primary outcomes are urinary pesticide metabolites in a 15% random subsample of participants and dietary diversity in all participants at 12 months post-baseline assessment. Both primary outcomes will be measured in (1) adult men ≥18 years old, (2) adult women ≥18 years old, and (3) children <38 months old at enrolment.

## Methods

### Aims

The primary aim of this study is to determine if a community-based agroecology programme implemented by the state government of Andhra Pradesh in India results in lower urinary pesticides and higher dietary diversity among adults and children in agricultural households, as compared to standard agricultural practices (primary outcomes).

Secondary aims include to determine whether this programme improves crop yields, household income, adult anthropometry, anaemia, glycaemia, kidney function, musculoskeletal pain, clinical symptoms, depressive symptoms, and women's empowerment, and child growth and development in the same adults and children (secondary outcomes).

In an a priori secondary analysis, data collected on compliance with APCNF practices will be used to estimate the per-protocol effect of APCNF on the outcomes.

### Setting

The study will be located in four (Anakapalli, Kurnool, Nandyal, and Visakhapatnam) of Andhra Pradesh's 26 districts selected by the research team to capture different agro-climatic zones. When the study was initiated, these geographical areas represented two districts, Visakhapatnam and Kurnool, but following district bifurcation in April 2022, these areas now

**Table 2. Comparison of two original districts in BLOOM (co-Benefits of Largescale Organic farming On huMan health) study to state- and national-level data in India.**

| | Kurnool (district) | Vishakhapatnam (district) | Andhra Pradesh (state) | India (national) |
|---|---|---|---|---|
| Total population[*] | 4.05 million | 2.29 million | 49 million | 1,210.2 million |
| Rural population (% total)[*] | 72% | 53% | 70% | 65% |
| Female literacy rate[†] | 57% | 70% | 67% | 72% |
| Total land area (hectares)[‡] | 1,765,800 | 1,116,100 | 16,297,000 | 297,319,000[*] |
| Cultivated land (% total)[‡] | 53% | 30% | 45% | 60%[*] |
| Irrigated land (% cultivated land)[‡] | 28% | 27% | 52% | 38%[*] |
| Top 5 crops in terms of production[‡] | Rice, jowar (sorghum), maize, red gram, and green gram | Rice, sugarcane, palm oil, betel leaves, and maize | Rice, maize, Bengal gram, jowar (sorghum), black gram | Sugarcane, rice, wheat, cotton, maize |
| Prevalence of stunting in children under 5 years[†] | 51% | 31% | 31% | 36% |
| Prevalence of anaemia in reproductive-aged women (15–49 years)[†] | 59% | 58% | 59% | 57% |
| Prevalence of obesity including overweight in reproductive-aged women (15–49 years)[†] | 29% | 24% | 36% | 24% |
| Prevalence of diabetes in reproductive-aged women (15–49 years)[†] | 15% | 17% | 20% | 14% |
| Prevalence of diabetes in men (15–49 years)[† §] | *21%* | *18%* | 22% | 16% |

[*] District and state data are from the Census of India 2011. National data are from the World Development Indicators (World Bank), accessed 12 July 2021.

[†] Data are from the National Family Health Survey 2019–2020 (NFHS-5).

[‡] Data are from Agricultural Statistics at a Glance 2019 (Andhra Pradesh and India), Directorate of Economics & Statistics, or district handbooks (2018 for Vishakhapatnam and 2019 for Kurnool).

[§] District-level sample sizes for men are small and should be interpreted with caution.

represent four districts. Given how recent district bifurcation was in the state, statistics are not available for the four new districts and so hereafter, the two original districts are described. The population of Kurnool is 4.05 million (72% rural) and Visakhapatnam is 2.29 million (53% rural) [76]. The female literacy rate is 57% in Kurnool and 70% in Visakhapatnam [77]. About half of the total geographical area of Kurnool is cultivated (937,278 out of 1,765,800 hectares) and 28% of cultivated land is irrigated (263,000 hectares) [78]. In Visakhapatnam, 30% is cultivated (339,759 out of 1,116,100 hectares) and 27% of cultivated land is irrigated (90,193 hectares) [79]. The top five crops in terms of production in Kurnool are rice, jowar (sorghum), maize, red gram, and green gram [78], and in Visakhapatnam, rice, sugarcane, palm oil, betel leaves, and maize [79]. With regards to nutritional and health outcomes, the prevalence of stunting in children under 5 years of age is 51% and 31% in Kurnool and Visakhapatnam, respectively, and the prevalence of anaemia in women is 59% and 58%, respectively [77]. The prevalence of obesity including overweight (body mass index [BMI] $\geq$25 kg/m$^2$) in women is 29% and 24% in Kurnool and Visakhapatnam, respectively, and the prevalence of diabetes is 15% and 17%, respectively [77]. These values are generally comparable to values at the state and national level (Table 2).

## Government intervention

A Template for Intervention Description and Replication checklist for population health and policy interventions [80] is provided in S1 Table. The intervention is the APCNF programme implemented by Rythu Sadhikara Samstha (RySS), a not-for-profit company established by the Department of Agriculture, Government of Andhra Pradesh (Fig 1). Farmers allocated to the

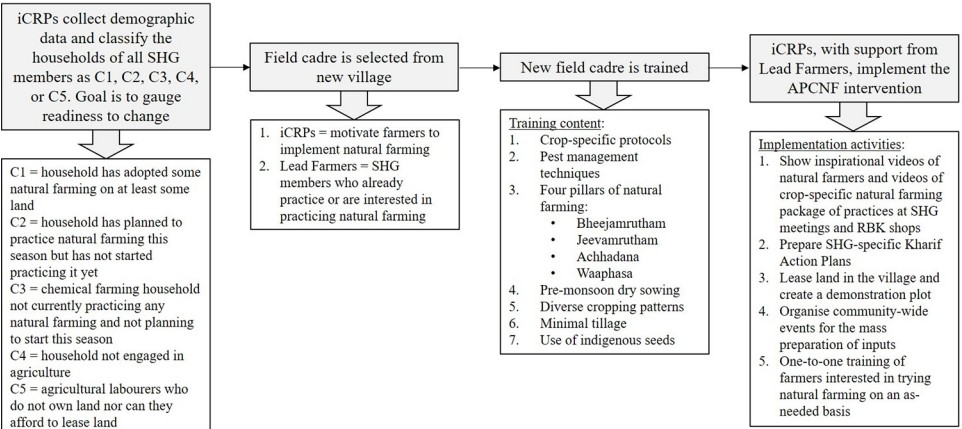

**Fig 1. Implementation process of the Andhra Pradesh Community-managed Natural Farming programme in India.** Abbreviations: APCNF, Andhra Pradesh Community-managed Natural Farming; iCRP, internal Community Resource Person; RBK, Rythu Bharosa Kendras; SHG, Self-Help Group; VO, Village Organisation.

APCNF group will receive training in APCNF practices. At the cluster level, the training is primarily provided by internal Community Resource Persons (iCRPs), who are long-term natural farming practitioners. iCRPs will be paid by RySS to live in a cluster and motivate and support farmers in adopting APCNF practices.

In terms of APCNF personnel training, after their initial training, the field cadre will attend district-level review meetings every month during which project progress will be discussed. Any new protocols for training will also be introduced during these monthly meetings. RySS also provides training modules on production, research, marketing, and institution building for dissemination in both print and digital formats. An app containing these materials is also under development.

Following randomisation, an iCRP will enter the clusters randomised to receive the intervention and engage with the Village Organisation and all of the women's self-help groups (SHGs) present in the village. The iCRPs will conduct regular trainings and meetings with SHGs, and, in parallel, initiate conversations and training directly with farmers. When farmers show an interest in adopting APCNF practices, trainings and demonstrations will happen with those individual farmers as well as in groups.

Women's SHGs will also support implementation. iCRPs will undertake training and capacity-building activities with the SHGs to identify lead farmers for the programme within the SHGs. All the lead farmers will be provided with continuous support such that they can further guide other SHG members to adopt APCNF practices. Each SHG consists of 10–15 members from the same community and living in the same neighbourhood. With respect to the APCNF programme, these SHGs are responsible for assessing input availability, facilitating input preparation, nominating lead farmers, and promoting APNCF practices.

The APCNF programme is therefore gender-sensitive. The gender component includes mobilisation of women's SHGs as mentioned above, as well as training female iCRPs, promoting the establishment of home gardens by women, and supporting women's decision-making in agriculture and control over the use of agricultural income.

With regards to APCNF programme content, in addition to adhering to zero synthetic chemical inputs, the APCNF programme emphasises the following four pillars: (1) microbial seed coating with cow dung- and urine-based formulations, (2) enhancing the soil microbiome by integrating cow dung and urine, and (3) cover cropping and mulching, which together

result in (4) greater soil humus, improved soil aeration, and water retention. The programme also includes pre-monsoon dry sowing (cover cropping), diverse cropping patterns, using botanical extracts for pest management, minimal tillage, and using indigenous seeds.

There will be no charge for the APCNF programme. There will also be no monetary incentives provided to farmers who participate in the APCNF programme. At the end of the study, the conventional practices group (i.e., control) will receive the APCNF training.

## Study logic model

A logic model [81] was developed by the evaluation team with inputs from the individuals leading the implementation of APCNF to understand programme activities and select indicators for evaluation (Fig 2). A logic model is a graphical representation of the way the programme is expected to work and links the inputs of the programme with the activities, outputs, outcomes, and impacts [82]. It does not include every detail about the programme but indicates the critically important aspects relevant for programme evaluation.

The BLOOM logic model consists of inputs, activities, outputs, outcomes (short-term and medium-term), and impacts. The first three components of the logic model are further divided into two groups–programme-related and evaluation-related. The former describes what is currently present with respect to the APCNF programme whereas the latter describes the additional work the evaluation team is proposing to conduct. The short-term outcomes are changes that the evaluation and implementation teams hypothesise will occur after 12 months of implementation of APCNF whereas the medium-term outcomes are expected after 24 months. Although the implementation team expects other changes due to the APCNF programme, we have chosen to present those which are agreed upon by both the evaluation and implementation teams.

## Study design

This study is a community-based, cluster-randomised controlled evaluation with two parallel groups: APCNF ('intervention') versus conventional practices ('control'). A total of 80 clusters will be randomly allocated 1:1 to intervention or control. The study will run for 48 months, including evaluation development and setup; baseline, midline, and endline assessments; intervention implementation (24 months); laboratory measurements of biomarkers; and statistical analysis. The main study assessments will be conducted during monsoon cultivation (July-December) with a brief survey also conducted after the monsoon harvest (January-February). The same random subset of households enroled at baseline from each cluster will be followed at 12 and 24 months post-baseline assessment.

## Selection of clusters and random allocation procedures

A list of clusters in which APCNF has not yet been implemented will be provided by RySS and the evaluation team will randomly select clusters for inclusion in the BLOOM study from this list using simple random sampling. By purposefully enrolling clusters where RySS has never before worked, we will minimise what has been described elsewhere as the '[NGO] reputation effect' [83]. In a previous randomised controlled trial in India, this reputation effect–meaning prior engagement with target communities–was shown to bias intervention effectiveness by at least 30% [83].

The study will be explained to elected representatives in the village as well as other stakeholders (e.g., community health workers and Anganwadi [government childcare centre] workers) and their permission sought. Given the narrow agricultural season window, clusters will be randomised to intervention or control after recruitment and enrolment, and the list shared

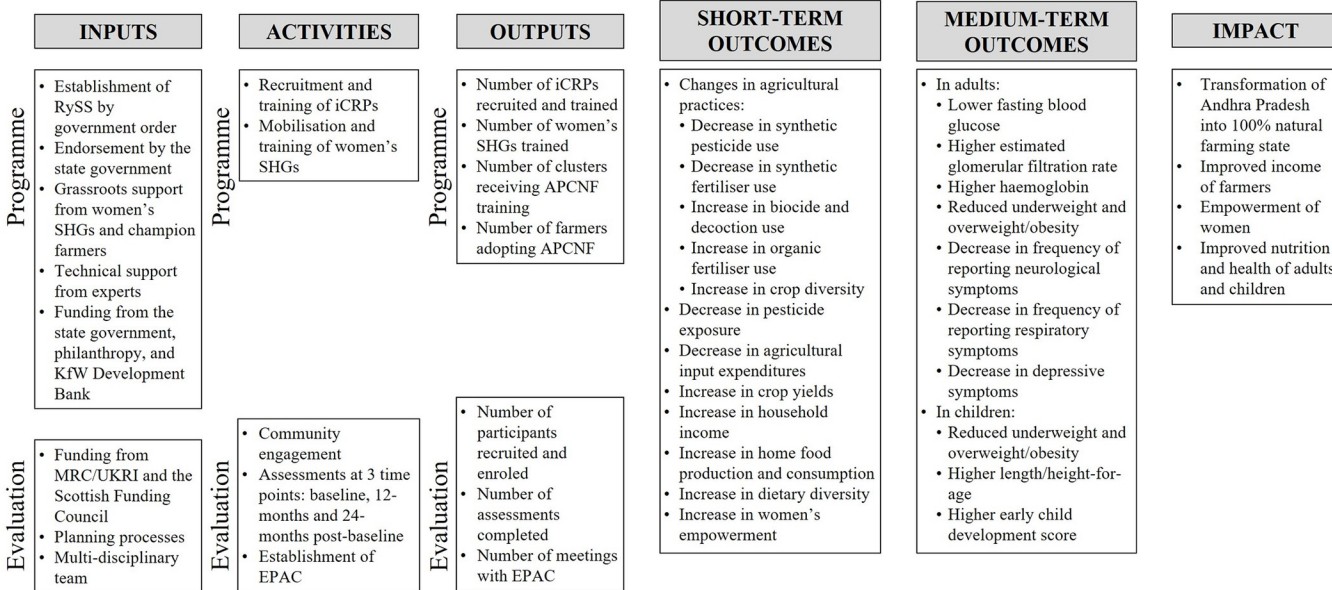

**Fig 2. The BLOOM (co-Benefits of Largescale Organic farming On huMan health) study logic model for the evaluation of the Andhra Pradesh Community-managed Natural Farming programme in India.** Abbreviations: EPAC, Evaluation and Policy Advisory Committee; iCRPs, internal Community Resource Persons; MRC, Medical Research Council; RySS, Rythu Sadhikara Samstha; SHGs, self-help groups; UKRI, UK Research and Innovation.

in confidence with RySS so that they can begin planning implementation activities for intervention clusters. No implementation activities will begin, and data collectors and participants will not be informed of the randomisation status, until the baseline assessment is complete. Randomisation will be stratified by original district (Kurnool and Vishakhapatnam).

## Selection of evaluation participants

The sample will be selected via random selection of 34 households per cluster from a roster of households in each cluster with a child <3 years of age. Rosters will be obtained by conducting a comprehensive household listing in each cluster based on lists provided by Anganwadi workers. In the event that a cluster has fewer than 34 eligible households, all eligible households will be enroled.

Three participants will be enroled from each household: an adult male, adult female, and child. The parents of the child, if eligible, will be enroled. Eligibility criteria are summarised in Table 3. If one or both parents of the child are not available or not eligible, we will enrol the oldest eligible male or female in the household. If more than one child is eligible in a given household, we will enrol the oldest eligible child.

Each participant will have data collected at three time points: baseline, 12 months, and 24 months (Fig 3). Data at each time point will be collected over the course of four waves. During wave 1, written informed consent will be obtained from participants, participants will be screened for eligibility, and individual questionnaires will be administered (demographics, tobacco and alcohol use, clinical history including medication use, and depressive symptoms). During wave 2, two urine samples and one fasting blood sample will be collected from up to 12 households per cluster, and dietary intake, infant and young child feeding, and time use will be assessed in all households (coincident with the urine sample collection). During wave 3, physical measurements will be taken (blood pressure, weight, and length/height), child development

**Table 3. Participant eligibility criteria for the BLOOM (co-Benefits of Largescale Organic farming On huMan health) study, a cluster-randomised controlled evaluation of Andhra Pradesh Community-managed Natural Farming in India.**

|  | Adults | Children |
|---|---|---|
| **Inclusion criteria** | 1. Household engaged in agriculture work defined as any one or more of the following: owning land, harvesting a crop in the past month regardless of land ownership, or earning a daily wage or contract-based wage for agricultural activities. This includes farmers, farm owners, farm workers, field workers, growers, harvesters, packers, graders and sorters, as well as agricultural pesticide handlers (mixers, loaders, cleaners and sprayers).<br>2. Aged ≥18 years. Age will be confirmed by directly viewing a government-issued document with the individual's date of birth.<br>3. Permanently reside in the selected household. 'Residence' will be defined as a group of people who eat from the same kitchen.<br>4. Have a child aged <38 months old who also resides in the household.<br>5. Willing to provide informed consent. | 1. Aged <38 months. Age will be confirmed by directly viewing a government-issued document with the child's date of birth.<br>2. Permanently reside in the selected household. 'Residence' will be defined as a group of people who eat from the same kitchen.<br>3. Parent or legal guardian willing to provide informed consent. |
| **Exclusion criteria** | 1. Plan to move permanently out of the study area in the next 12 months.<br>2. Visible disabilities (e.g., physical malformations or genetic syndromes) that prohibit participation.<br>3. Primary language other than Telugu.<br>4. Already enroled in a different research study.<br>5. Unwilling to provide informed consent. | 1. Visible disabilities (e.g., physical malformations or genetic syndromes) that prohibit participation.<br>2. Already enroled in a different research study.<br>3. Parent or legal guardian unwilling to provide informed consent. |

will be assessed, and individual questionnaires will be administered (household economics and women's empowerment). Finally, during wave 4, individual questionnaires will be administered (clinical symptoms and musculoskeletal pain, agricultural practices, and pesticide use). Household questionnaires (e.g., household economics and agricultural practices) will be administered to the head of household and the child development and infant and young child feeding questionnaires to the primary caregiver. A health report will be given to participants during waves 2 and 3. The assessments for each wave will last no longer than 2 hours. In the event that previously undiagnosed health conditions are identified (e.g., hypertension, anaemia, type 2 diabetes, severe wasting, or suicidal ideation), participants will be referred to the nearest health centre.

## Blinding

It is not possible to blind participants to the intervention. Laboratory technicians and statistical analysts will be blinded. Field staff who collect the data will not be involved in intervention implementation.

## Primary outcomes

1. Urinary pesticide metabolites at 12 months in a 15% random subsample of adult men enroled in the study. Six common dialkylphosphate (DAP) metabolites of organophosphorus insecticides will be measured in an equal-volume composite urine sample consisting of two spot urine samples collected within a one-week period. Compositing samples reduces

| | Enrolment | | | Allocation | Post-allocation | | | | | |
|---|---|---|---|---|---|---|---|---|---|---|
| TIMEPOINT | $-t_1$ | | | 0 | $t_1$ | | | $t_2$ | | |
| LEVEL | M | F | C | Village | M | F | C | M | F | C |
| **ENROLMENT:** | | | | | | | | | | |
| Eligibility screen | X | X | X | | | | | | | |
| Informed consent | X | X | X | | | | | | | |
| Allocation | | | | X | | | | | | |
| **INTERVENTION:** | | | | | | | | | | |
| Government-implemented community-managed natural farming | | | | | ←——————————————→ | | | | | |
| **ASSESSMENTS:** | | | | | | | | | | |
| *Surveys* | | | | | | | | | | |
| Demographics | X | X | | | X | X | | X | X | |
| Tobacco and alcohol use | X | X | | | X | X | | X | X | |
| Clinical history | X | X | | | X | X | | X | X | |
| Depressive symptoms | X | X | | | X | X | | X | X | |
| Time use | X | X | | | X | X | | X | X | |
| Dietary intake | X | X | | | X | X | | X | X | |
| Infant and young child feeding | | | X | | | | X | | | X |
| Caregiver-Reported Early Child Development | | | X | | | | X | | | X |
| Household economics | X | | | | X | | | X | | |
| Women's empowerment | X | X | | | X | X | | X | X | |
| Agricultural practices | X | | | | X | | | X | | |
| Pesticide use | X | X | | | X | X | | X | X | |
| Musculoskeletal pain | X | X | | | X | X | | X | X | |
| Clinical symptoms | X | X | | | X | X | | X | X | |
| *Physical Measurements* | | | | | | | | | | |
| Anthropometry | X | X | X | | X | X | X | X | X | X |
| Blood Pressure | X | X | | | X | X | | X | X | |
| *Biosample Collection* | | | | | | | | | | |
| Urine | X | X | X | | X | X | X | X | X | X |
| Blood | X | X | | | X | X | | X | X | |

**Fig 3. Schedule of enrolment, intervention, and assessments for the BLOOM (co-Benefits of Largescale Organic farming On huMan health) study, a cluster-randomised controlled evaluation of Andhra Pradesh Community-managed Natural Farming in India.** 'M' is adult male, 'F' is adult female, and 'C' is child in enrolled households.

intra-person variability. The six DAPs will include: dimethylphosphate (DMP), dimethylthiophosphate (DMTP), dimethyldithiophosphate (DMDTP), diethylphosphate (DEP), diethylthiophosphate (DETP), and diethyldithiophosphate (DEDTP). Together, these six DAPs cover approximately 77% of organophosphorus insecticides produced in India (Table 4) [11]. Total DAP molar weight ($\Sigma$DAPs) will be calculated by multiplying the concentration of each of the six DAPs by the molar weights of the respective DAP and summing them. Specific gravity-adjusted $\Sigma$DAPs will be analysed as the primary outcome. Specific gravity will be measured in the composite urine sample using a handheld refractometer (Atago PAL-10S). Urine aliquots will be shipped on dry ice to the laboratory for analysis in random sequence using gas chromatography-mass spectrometry (GC-MS) [84]. Laboratory technicians will be blinded to the randomisation status of samples.

**Table 4. Top organophosphorus insecticides produced in India and their potential dialkyl phosphate metabolites.**

|  | DMP | DMTP | DMDTP | DEP | DETP | DEDTP |
|---|---|---|---|---|---|---|
| Acephate | X | X |  |  |  |  |
| Chlorpyriphos |  |  |  | X | X |  |
| Dimethoate | X | X | X |  |  |  |
| Ethion |  |  |  | X | X | X |
| Malathion | X | X | X |  |  |  |
| Monocrotophos | X |  |  |  |  |  |
| Temephos | X | X |  |  |  |  |

Abbreviations: dimethylphosphate (DMP), dimethylthiophosphate (DMTP), dimethyldithiophosphate (DMDTP), diethylphosphate (DEP), diethylthiophosphate (DETP), and diethyldithiophosphate (DEDTP).

2. Urinary pesticide metabolites at 12 months in a 15% random subsample of adult women enroled in the study. The same method, as described above for adult men, will be used to assess specific gravity-adjusted ΣDAPs in adult women.

3. Urinary pesticide metabolites at 12 months in a 15% random subsample of children enroled in the study. The same method, as described above for adult men, will be used to assess specific gravity-adjusted ΣDAPs in children.

4. Dietary diversity score at 12 months in all adult men enroled in the study. Dietary intake of adults will be assessed using a single 24-hour dietary recall. Participants will be taken through the previous day from morning to evening and asked to recall all foods and beverages consumed and the portion size. A standard set of locally appropriate plates, bowls, and cups and show cards will be used to aid portion size estimation. Dietary diversity will be calculated using the FAO's Minimum Dietary Diversity for Women (MDD-W) [85]. Ten food groups are included: (1) starchy staples (rice, wheat, and potatoes), (2) legumes and pulses, (3) nuts and seeds, (4) dark green leafy vegetables, (5) vitamin-A rich fruits and vegetables, (6) other vegetables, (7) other fruits, (8) dairy products, (9) eggs, and (10) flesh foods (meat, poultry, and fish). For analysis, men who consume a food group on the previous day will be assigned a value of 1 and those who do not will be assigned a value of 0 and the values across these ten food groups will be summed in a dietary diversity score. Thus, the dietary diversity score can range from 0 to 10 with 10 representing maximum dietary diversity.

5. Dietary diversity score at 12 months in all adult women enroled in the study. The same method, as described above for adult men, will be used to assess dietary diversity in adult women.

6. Dietary diversity score at 12 months in all children enrolled in the study. Dietary intake of children will be assessed using the standard WHO Infant and Young Child Feeding practices survey [86], adapted to the local context and administered to the primary caregiver of the child. Dietary diversity will be calculated using the WHO's dietary diversity score (DDS). Whilst this was originally developed for children 6–23 months old, DDS can serve as an adequate proxy for micronutrient intake in children 24–59 months as well [87]. Eight food groups are included: (1) breast milk, (2) grains, roots, tubers, and plantains, (3) pulses, nuts, and seeds, (4) vitamin-A rich fruits and vegetables, (5) other fruits and vegetables, (6) dairy products, (7) eggs, and (8) flesh foods (meat, poultry, and fish). For analysis, children who consume a food group on the previous day will be assigned a value of 1 and those who do not will be assigned a value of 0 and the values across these eight food groups will be summed. Thus, DDS can range from 0 to 8 with 8 representing maximum dietary diversity.

## Secondary outcomes

Urinary pesticide metabolites and dietary diversity at 24 months will be assessed as secondary outcomes. In addition, the following secondary outcomes will be measured at 12 and 24 months post-randomisation.

1. Fasting plasma glucose. A trained phlebotomist will collect an overnight fasting blood sample. Once collected, the sample will be placed in an insulated box on ice packs and transported to the laboratory for processing and same-day analysis of glucose by the hexokinase method [88].

2. Estimated glomerular filtration rate will be calculated using the Modification of Diet in Renal Disease Study equation [89]. Serum creatinine will be measured using the Jaffe's method, calibrated using Isotope Dilution Mass Spectroscopy (IDMS) traceable-creatinine. Blood pressure will be measured in order to differentiate between chronic kidney disease (CKD) and CKD of unknown aetiology (CKDu). Systolic and diastolic blood pressure will be measured in triplicate with at least 2 minutes resting between repeat measurements using an automatic blood pressure monitor (Omron JPN1) after leaving the person rested and seated for at least 5 minutes before taking the first measurement. The average of the second and third measurements will be used in the analysis.

3. Urine protein-to-creatinine ratio will be measured using the first sample of two spot urine samples collected from the adult participants. Urine protein will be measured using the pyrogallol red-molybdate method. Urine creatinine will be measured using the Jaffe's method.

4. Adult nutritional status. Weight and height of adults will be measured based on the National Health and Nutrition Examination Survey protocol [90] using electronic scales (Seca 874) and portable stadiometers (Seca 213). Adult BMI will be calculated as weight (kg) divided by height-squared ($m^2$) and categorised as underweight (BMI $<18.5$ kg/$m^2$), normal weight (BMI 18.5 to $<25$ kg/$m^2$), and obesity including overweight (BMI $\geq25$ kg/$m^2$) [91]. Haemoglobin will be measured in venous blood samples using photometry. Anaemia will be defined as haemoglobin $<12$ g/dL in non-pregnant women, haemoglobin $<11$ g/dL in pregnant women, and haemoglobin $<13$ g/dL in men, according to the WHO guidelines [92].

5. Self-reported musculoskeletal pain and clinical symptoms. Musculoskeletal pain and clinical symptoms in the past 3 months will also be queried. Symptoms will include itchy skin, rash on skin, dry/cracking skin, blisters on the skin, fatigue or weakness, headache, dizziness, eye irritation or watering, pain in the eye, redness in the eye, blurry vision, chest pain / tightness, difficulty breathing, and digestive problems (e.g., nausea, vomiting, stomach cramp, diarrhoea). The list of symptoms was adapted from a survey on pesticide use in Thailand [93].

6. Depressive symptoms. Depressive symptoms will be assessed using the Patient Health Questionnaire (PHQ, 9 items) [94], which has been used previously in this context [95, 96]. The questionnaire asks participants how often (not at all, several days, more than half of the days, or nearly every day) over the last 2 weeks they have been bothered by problems such as having little interest or pleasure in doing things; having trouble concentrating on things; and thoughts that they would be better off dead or of hurting themselves in some way. Given the sensitive nature of these questions, the survey will be completed in a private area and data collectors will confirm with the participant that they feel comfortable before beginning the survey.

7. Women's Empowerment. Women's empowerment will be measured using the abbreviated Women's Empowerment in Agriculture Index (A-WEAI) [97] and the abbreviated Women's Empowerment in Nutrition Index (A-WENI) [98]. The A-WEAI assesses women's empowerment and inclusion in agriculture. It covers five domains: production, resources, income, leadership, and time. The A-WENI assesses women's nutritional empowerment or women's capacity to be healthy and well fed. It covers seven domains: food knowledge, food resources, food agency, health knowledge, health resources, health agency, and institutions. For both A-WEAI and A-WENI, domain-specific scores and total empowerment scores will be calculated.

8. Child growth (length/height-for-age z-score). Child length ($\leq$2 years) and height (>2 years) will be measured using a portable length board (Seca 417) or stadiometer (Seca 213). Child stunting will be defined as length/height-for-age z-score < -2 SD below the median z-score based on WHO Child Growth Standards [99].

9. Child development. At baseline, the Caregiver-Reported Early Development Instruments (CREDI), will be used to assess child development in children <38 months of age [100]. At 12 and 24 months post-randomisation, we will use the Parents' Evaluation of Developmental Status:Developmental Milestones (PEDS:DM) to assess child development [101]. Both CREDI and PEDS:DM assesses motor, cognitive, language, and socio-emotional development based on caregiver report. We will assess raw scores, sample-standardised scores, and norm-referenced standardised scores.

10. Household crop yield will be collected via self-report using questions adapted from the Indian National Sample Survey Office's (NSSO) Situation Assessment Survey of Agricultural Households [102].

11. Household income, expenditures, and debt will be collected via self-report using questions adapted from the NSSO Situation Assessment Survey of Agricultural Households [102].

## Additional assessments

Demographic and socio-economic data will be collected at baseline for all participants using questions adapted from the 5th round of the Indian National Family Health Survey (NFHS-5) [5]. Specific variables to be assessed include age, sex, marital status, educational attainment, occupation, caste, cooking fuel, source of drinking water, toilet facility, household construction materials, and asset ownership including livestock. A household wealth index will be calculated from these variables using principal components analysis.

Information on cropping pattern, land cultivated, land irrigated, and land owned, in both Kharif (monsoon season) and Rabi (winter season), will be collected using questions adapted from the Indian Agriculture Census [103] and NSSO Situation Assessment Survey of Agricultural Households [102]. Farmer estimates will be used to estimate crop yields, quantity sold, where it was sold, and the sale value. This survey will also include questions on chemical input use, pesticide storage, and all expenses relating to crop production in the past 12 months (seeds, soil, fertilisers, manure, pesticides, diesel, electricity, human labour, animal labour, irrigation, minor repairs and maintenances, machinery hire, and lease rent for land). Detailed information will be collected from all adult participants on years working in agriculture; how many days per week and hours per day engaged in agricultural work; and which specific agricultural activities are undertaken. For those reporting activities relating to pesticides (e.g., mixing, loading, and/or application), additional information on chemicals used, application rate, method of mixing, method of application, use of personal protective measures and/or

equipment, and personal hygiene practices will be collected. This survey will also query domestic use of pesticides for insect control. Questions are adapted from a survey on pesticide use in Thailand [93].

We will measure key practices emphasised by the APCNF programme in order to quantify fidelity. Indicators, as provided by RySS, will include poly cropping, inter cropping, multilayer cropping, border crops, trap crops, fruit trees, pre-monsoon dry sowing, cattle grazing on pre-monsoon dry sowed crop, 365 days green cover, indigenous seeds, Beejamrutham seed coating, minimal tillage, mulching, manual weeding, mechanical weeding, irrigation, botanical inputs for pest management, pheromone and sticky traps for pest management, use of Jeevamruth, and use of cattle manure.

Self-reported medical history and current medication use will be assessed for all participants. Diseases will include diabetes, hypertension, high blood cholesterol, heart disease/angina/heart attack/stroke, chronic kidney disease, kidney stones, asthma/chronic obstructive pulmonary disease/emphysema, cancer, and cataracts. We will also collect information on pesticide poisoning (both intentional and unintentional). Information will be collected from women on their number of pregnancies and number of live births.

## Collection and storage of biological specimens

Urine and blood samples will be collected at three time points from adults, and urine at three time points for children in up to 12 households per cluster. Sample aliquots will be stored at -80˚C in New Delhi, India at the Public Health Foundation of India's biospecimens repository.

## Power calculations

Power calculations were conducted using PASS 2019, v19.0.6 (NCSS, Kaysville, Utah, USA), accounting for the cluster-randomised design. Whilst hypotheses for the two primary outcomes (urinary pesticides and dietary diversity) were independent, in order to be conservative, a Bonferroni correction was applied (0.05/2, which is the number of primary outcomes) resulting in a two-sided alpha level of 0.025. It is expected that approximately 20% of the baseline sample will be loss-to-follow-up at 12 months. Two previous surveys in Andhra Pradesh by study team members had much lower loss to follow-up rates [2% from 2019 to 2020 in the SHEFS (Sustainable and Healthy Food Systems) nutrition survey (unpublished) and 10% for the mWellcare trial [104] but a higher rate was anticipated for the BLOOM study due to COVID-19.

Power for the primary outcomes is given in Table 5, which summarises the minimal detectable difference in mean levels associated with 80% power, a type I error rate alpha of 0.025, an intra-cluster correlation of 0.02, and 20% loss to follow-up. For ΣDAPs power calculations are shown for 150 per arm (a 15% subsample). Results in Table 5 confirm that the study is adequately power to detect a smaller difference than has been previously reported in the literature for these outcomes.

## Data collection and management

Study data will be collected and managed using REDCap (Research Electronic Data Capture) hosted at the Public Health Foundation of India [109, 110]. REDCap is a secure, web-based software platform designed to support data capture for research studies. Data will be collected using an offline mobile app on a password-protected tablet and then synced with REDCap's secure, web-based software platform. A de-identified dataset that does not contain any information that could lead to participant identification will be made publicly available on the Central Research Data Repository of the Public Health Foundation of India and Edinburgh DataShare upon publication of the primary outcomes.

**Table 5. Minimum detectable mean difference, with 80% power, alpha = 0.025, and intra-cluster correlation of 0.02.**

|  | Standard Deviation for Mean Change in Outcome | Sample Size per Group | Minimum Detectable Mean Difference | Previous Study Estimate (95% Confidence Interval) |
|---|---|---|---|---|
| Change in mean ∑DAPs, adult men | 111 μmol/l[*] | 150 | 39.6 μmol/l (~55% of estimated baseline levels[*]) | -88% (not reported)[†] |
| Change in mean ∑DAPs, adult women | 111 μmol/l[*] | 150 | 39.6 μmol/l (~55% of estimated baseline levels[*]) | -88% (not reported)[†] |
| Change in mean ∑DAPs, children | 17.3 nmol/l[‡] | 150 | 6.17 nmol/l (~9.7% of estimated baseline levels[‡]) | -42.7% (-76.3–38.7)[‡] -39.9% (-58.6, -12.6)[§] |
| Change in mean dietary diversity score, adult women | 1.57[¶] | 1000 | 0.29 | 0.36 (0.03–0.69)[¶] |
| Change in mean dietary diversity score, children | 1.40[#] | 1000 | 0.26 | 0.39 (0.13–0.64)[#] |

[*] From an observational study of adult agricultural labourers in North Carolina [105].

[†] From an observational cross-sectional study comparing organic and conventional vegetable farmers in Thailand [106]. Difference is for urinary levels of 3,5,6-trichloro-2-pyridinol, a metabolite of chlorpyrifos, a specific organophosphate.

[‡] Percent reduction in ∑DAPs reported for children under 4 years old participating in a home-based educational intervention to reduce take-home pesticides by agricultural labourers in California, USA [107]. Change is from baseline to 3 months.

[§] Percent reduction in ∑DAPs reported for children 3–6 years old living in urban or agricultural communities in California, USA, who were fed organic food for 4 days [108]. Change is between the control period (9 days) and intervention period (7 days).

[¶] From a cluster-randomised controlled trial of a nutrition-sensitive agroecology intervention in Tanzania for which a secondary outcome was minimum dietary diversity among women [30]. The estimates used in these power calculations are from the 12-month follow-up visit. Dietary diversity was assessed using the FAO's minimum dietary diversity score for women, the same as BLOOM. The standard deviation for mean change in outcome is estimated from the control group.

[#] From a cluster-randomised controlled trial of a nutrition-sensitive agroecology intervention in Tanzania for which the primary outcome was dietary diversity among children aged <1 year at baseline [30]. The estimates used in these power calculations are from the 12-month follow-up visit. Dietary diversity was assessed using the older version of the WHO's minimum dietary diversity score that did not include breastmilk and so was out of a total of seven food groups instead of eight. The standard deviation for mean change in outcome is estimated from the control group.

## Data quality control

Data collectors will complete a 5-day training and must pass an examination before being certified for field work. Throughout the data collection period, field supervisors will directly observe data collectors on a regular basis. The study will employ the use of Standard Operating Procedures (SOPs), which will document, in detail, all methods used to generate the data. With written consent, audio-video recordings and photographs will be taken during the study visits for the purpose of improving the quality of data collected by the study team. All Case Report Forms (CRFs) will include an entry for the person who collected the data to provide their initials.

Regular Quality Control (QC) reports will be run by the data manager to identify entry errors and missing data. To help ensure data completeness, data collectors will be prompted to go back and complete any missing values when they attempt to submit an incomplete CRF. Skip patterns will be automatically implemented in REDCap CRFs. Wherever possible, maximum and minimum limits will be implemented in REDCap CRFs (e.g., if asked to report number of days in the past week, values >7 will not be permitted).

## Statistical analysis plan

The main analyses will be intent-to-treat, i.e., participants will be analysed as assigned, regardless of their fidelity to APCNF practices. Herein the analyses for primary outcomes are described. Analyses for secondary outcomes will be analogous. Separate models will be run for each of the primary outcomes. All primary outcomes are continuous. General estimating equation (GEE) models with robust standard errors will be used to account for within-participant

correlations [111]. Models will account for clustering within villages. The models will take the form of:

$$Y_{i,t} = \beta_0 + \beta_1 Group + \beta_2 Time + \beta_3 Group * Time + \beta_4 District + \varepsilon_{i,t} \qquad (1)$$

where group is APCNF or control, time is baseline or 12 months, and district is Kurnool or Visakhapatnam (to account for randomisation stratification). $\beta_3$ represents the difference in the change from baseline to 12 months in the APCNF group and the change from baseline to 12 months in the control group (e.g., difference in difference).

We will evaluate whether missing data are differential between groups using t-tests. We will assume missing data to be missing at random and conduct a complete-case analysis. Inverse probability weighting will be used to account for loss to follow-up. We will run adjusted models to address potential baseline imbalance across clusters and improve precision. Covariates will be selected based on p-values for differences across groups.

## Ethical oversight and study governance

The study protocol will be reviewed and approved by two ethics committees: (1) the Public Health Foundation of India's Institutional Ethics Committee and (2) the University of Edinburgh's Human Ethical Review Committee. Written informed consent will be obtained from all participants prior to conducting any assessments. The study is sponsored by the Academic & Clinical Central Office for Research & Development (ACCORD) at the University of Edinburgh, which provides oversight for clinical research conducted in the UK and oversees. The study is funded by the Medical Research Council/UK Research and Innovation and the Scottish Funding Council. The study sponsor and funders did not play a role in the study design and will not play a role in the collection, management, analysis, or interpretation of data; writing of the final report; or decision to submit the final report for publication.

The implementation of the BLOOM study will be overseen by a Study Management Committee (SMC), which meets at least monthly via video conference call. The study will also convene a Steering Committee with members who are independent from the study evaluation team. The Steering Committee will meet twice annually via video conference call. Their responsibilities, outlined in a Charter, will include: providing expert oversight of the study; monitoring recruitment rates; reviewing regular reports of the study from the SMC; assessing the impact and relevance of any accumulating external evidence; monitoring completion of CRFs; monitoring follow-up rates; commenting on any proposals by the SMC concerning any change to the design of the study, including additional ancillary studies; and overseeing the timely reporting of study results.

In addition to the SMC and Steering Committee, an Evaluation and Policy Advisory Committee (EPAC) will be formed with members from relevant state departments in Andhra Pradesh (health, agriculture, women and child development, education, and tribal welfare), the evaluation team, and the implementation team. The role of EPAC will primarily be to provide recommendations and connections to relevant initiatives and strategies at the state and national levels. They will also support dissemination of the research findings.

## Process evaluation

A three-part process evaluation will be conducted in accordance with Medical Research Council guidance for process evaluations of complex interventions [112]. First, implementation will be assessed including fidelity, dose, adaptations, and reach of the programme and how the programme was implemented (i.e., virtually versus face-to-face, group versus individual). Barriers and facilitators to implementation will also be assessed. Problems in implementation will be communicated to RySS as and when they appear.

Second, contextual factors that may influence the delivery of the programme or the study outcomes will be systematically recorded in all BLOOM villages. These factors will include other government programmes implemented in the village, any NGOs operating in the village and their activities, rainfall, and pest outbreaks. Third, mechanisms of impact including farmers' adherence to the programme, acceptability of natural farming practices, and barriers and facilitators to adoption will be assessed.

Data will be collected using a combination of quantitative surveys, diaries, qualitative in-depth interviews with implementers and farmers, direct observations of trainings, and secondary data sources (e.g., routine government monitoring data on rainfall and pest outbreaks).

## Discussion

### Study implications

The BLOOM study will provide robust evidence of the impact of a large-scale transformational government-implemented agroecology programme on pesticide exposure and dietary diversity in agricultural households in rural Andhra Pradesh, India. It will also provide the first evidence of the health co-benefits of adopting agroecology, inclusive of malnourishment and common chronic diseases including type 2 diabetes and kidney disease, as well as early child development. In working closely with the government organisation responsible for implementation from the outset of the study, the study team will help to ensure that the study's outcomes are priorities for the government and that the findings are rapidly disseminated and integrated into agriculture and public health policy in India. Whilst fully recognising that APCNF is unlikely to be the best solution in all contexts, this research study will provide evidence regarding its effectiveness in India with direct generalisation to other countries in South Asia and Africa considering adopting the approach [e.g., Nepal and Rwanda [113]].

This will be the first evaluation of a programme aimed at eliminating pesticide exposure among farmers in a low- or middle-income country to measure biomarkers of pesticide exposure. All previous studies in these contexts have relied on self-reported pesticide use. For example, our cross-sectional pilot study in Kurnool district, Andhra Pradesh, conducted between August and November 2020 found that APCNF farmers were 34% less likely to use synthetic pesticides compared to conventional farmers [12]. One cross-sectional comparison of n = 8 pesticide-free vegetable farmers and n = 11 conventional vegetable farmers in Chiang Mai, Thailand, found substantial reductions in urinary pesticide biomarkers [106].

Given that agroecology, including APCNF [114], explicitly promotes the diversification of crops cultivated, and crop diversity is positively associated with dietary diversity [114], one might expect agroecology to have an impact on dietary diversity. However, to the best of our knowledge, only one previous study, in Tanzania, has evaluated the impact of an agroecology programme on dietary diversity, finding significant improvements in child and women's dietary diversity [30].

Thus, the BLOOM study will fill an important evidence gap relating to the human health co-benefits of sustainable agriculture. Given the wide breadth of outcomes evaluated in BLOOM, the study has the potential to inform the design of future programmes and interventions regardless of main outcomes of interest. Considering that agroecology is increasingly recognised as the way forward in agriculture interventions/programmes, this is especially relevant.

### Study strengths and limitations

The BLOOM study is led by a multi-disciplinary team of researchers and leverages a scientifically rigorous study design, employing cluster randomisation, which will strengthen causal

inferences. The research questions were informed through discussions with policymakers and the government implementing agency, and government stakeholders have been involved since its inception. This is a randomised controlled evaluation in four districts of one state in south India and thus will not necessarily be generalizable to all districts in Andhra Pradesh nor all states in India. Nonetheless, these districts are relatively comparable to state- and national-level demographic and health characteristics (Table 2). The primary outcome of ∑DAPs will capture changes in organophosphate pesticides but not all pesticides. Our preliminary findings from Kurnool indicated that organophosphate pesticides were the most commonly reported pesticides used by farmers and the top-selling pesticides at retail shops [12]. However, if no change in ∑DAPs is observed, it will not be possible to make any conclusions about changes in other classes of pesticides such as pyrethroids, carbamates, neonicotinoids, or fungicides. Additional limitations include the use of self-report to assess dietary intake and APCNF fidelity.

## Study status

Recruitment and enrolment began in June 2022. Baseline assessments will be completed by February 2023. Programme implementation by the government will begin in February 2023 in clusters allocated to the intervention group.

## Supporting information

**S1 Table. Template for Intervention Description and Replication (TIDieR) checklist for population health and policy interventions.**
(DOCX)

## Acknowledgments

The BLOOM Study is possible due to a collaboration with RySS, the implementing organisation for APCNF. We would especially like to thank T. Vijay Kumar, IAS (Rtd) Ex-Officio Special Chief Secretary to Government of Andhra Pradesh for Agriculture & Cooperation and Executive Vice Chairman at RySS, and Lakshmi Durga Chava, Senior Consultant (Health and Nutrition), RySS. Other institutional partners include Ross Lifescience, Qpath Labs, Emory University, and Stanford University. We are grateful for the administrative support provided by Clarinda Brown, Senior Project Manager at the University of Edinburgh. We would also like to thank Niti Gupta, Research Coordinator at the Centre for Chronic Disease Control for her inputs on the APCNF programme implementation. We would like to acknowledge the helpful feedback of the BLOOM Steering Committee: Dr. Hemalatha R (Chair),Prof. Vivekanand Jha, Prof. Laura Gray, Dr. Keith Tyrell, Dr. RV Bhavani, and Ms. Anne-Sophie Poisot. Finally, we would like to thank Marianne (Vicky) Santoso, Rachel Bezner Kerr, Sera Young, and Thomas Arcury who generously provided data to inform our power calculations for BLOOM.

## Author Contributions

**Conceptualization:** Lindsay M. Jaacks, Michael Eddleston, Nikhil Srinivasapura Venkateshmurthy, Poornima Prabhakaran.

**Data curation:** Lindsay M. Jaacks, Sheril Rajan.

**Formal analysis:** Lindsay M. Jaacks.

**Funding acquisition:** Lindsay M. Jaacks, Poornima Prabhakaran.

**Investigation:** Lindsay M. Jaacks, Sheril Rajan, Bharath Yandrapu, Nikhil Srinivasapura Venkateshmurthy.

**Methodology:** Lindsay M. Jaacks, Lilia Bliznashka, Peter Craig, Michael Eddleston, Alfred Gathorne-Hardy, Ranjit Kumar, Sailesh Mohan, John Norrie, Aditi Roy, Bharath Yandrapu, Nikhil Srinivasapura Venkateshmurthy, Poornima Prabhakaran.

**Project administration:** Lindsay M. Jaacks, Bharath Yandrapu, Nikhil Srinivasapura Venkateshmurthy, Poornima Prabhakaran.

**Supervision:** Lindsay M. Jaacks, Bharath Yandrapu, Nikhil Srinivasapura Venkateshmurthy, Poornima Prabhakaran.

**Validation:** Lindsay M. Jaacks.

**Visualization:** Lindsay M. Jaacks.

**Writing – original draft:** Lindsay M. Jaacks.

**Writing – review & editing:** Lilia Bliznashka, Peter Craig, Michael Eddleston, Alfred Gathorne-Hardy, Ranjit Kumar, Sailesh Mohan, John Norrie, Sheril Rajan, Aditi Roy, Bharath Yandrapu, Nikhil Srinivasapura Venkateshmurthy, Poornima Prabhakaran.

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
