## [Decision Letter · Decision Letter 0]

1 Dec 2022

PONE-D-22-22524Co-Benefits of Largescale Organic farming On huMan health (BLOOM): Protocol for a cluster-randomised controlled evaluation of the Andhra Pradesh Community-managed Natural Farming programme in IndiaPLOS ONE

Dear Dr. Jaacks,

Thank you for submitting your manuscript to PLOS ONE. After careful consideration, we feel that it has merit but does not fully meet PLOS ONE’s publication criteria as it currently stands. Therefore, we invite you to submit a revised version of the manuscript that addresses the points raised during the review process. In particular, the reviewer is overall positive but Figure 1 needs clarification.

We look forward to receiving your revised manuscript.

Kind regards,

Arnab K. Basu

Academic Editor

PLOS ONE

Journal Requirements:

Reviewers' comments:

Reviewer's Responses to Questions

**Comments to the Author**

1. Does the manuscript provide a valid rationale for the proposed study, with clearly identified and justified research questions?

Reviewer #1: Yes

2. Is the protocol technically sound and planned in a manner that will lead to a meaningful outcome and allow testing the stated hypotheses?

Reviewer #1: Yes

3. Is the methodology feasible and described in sufficient detail to allow the work to be replicable?

Reviewer #1: Yes

4. Have the authors described where all data underlying the findings will be made available when the study is complete?

Reviewer #1: Yes

5. Is the manuscript presented in an intelligible fashion and written in standard English?

Reviewer #1: No

6. Review Comments to the Author

You may also provide optional suggestions and comments to authors that they might find helpful in planning their study.

Reviewer #1: I believe that the proposed BLOOM study is would be a very timely contribution to the debates around zero-budget natural farming (ZBNF) in India. While ZBNF has gotten great traction of late as a potential game-changer through food production increase without chemical fertilizer use to protect soil health. This is something which is high on the policy agenda, but there is little empirical evidence around the issue. Critics, infact, argue that ZBNF may not be a great idea as fertilizers in moderate quantity are essential for agricultural productivity increase. This study can therefore shed more light on this debate be a seminal contribution on this topic, which is increasingly getting policy traction in other areas of the world as well.

My concerns with the proposed study are as follows:

1. The agro-ecological interventions need to be spelt out better. There is a lot going on in terms of involving NGOs, encouraging women SHGs, and other community members to increase adoption. Each of these need to clearly explained.

2. While the agricultural impacts are easily understood, how (and why) the intervention is going to affect food consumption pattern is difficult to discern here. Is it through income or diversification of agriculture? I do not see a discussion on that.

3. Similarly, I am not able to understand the metrics chosen for health outcomes and the logic behind them. Also, is this a reasonable enough time to measure changes in health? Some change in health status could also emerge from the changing labor requirements post the intervention. How is that accounted for?

4. When NGOs are embedded in implementing an RCT, as it is proposed here, there is likely to be a higher take up of the programs creating a biased impact. I suspect, it is likely to be a case here. For reference, see Faraz Usmani, Marc Jeuland, Subhrendu K. Pattanayak; NGOs and the Effectiveness of Interventions. The Review of Economics and Statistics 2022.

7. PLOS authors have the option to publish the peer review history of their article (what does this mean?). If published, this will include your full peer review and any attached files.

Reviewer #1: **Yes: **Andaleeb Rahman

---

## [Author Response · Author response to Decision Letter 0]

9 Jan 2023

See attachment for formatted table of point-by-point responses.

1. I believe that the proposed BLOOM study is would be a very timely contribution to the debates around zero-budget natural farming (ZBNF) in India. While ZBNF has gotten great traction of late as a potential game-changer through food production increase without chemical fertilizer use to protect soil health. This is something which is high on the policy agenda, but there is little empirical evidence around the issue. Critics, infact, argue that ZBNF may not be a great idea as fertilizers in moderate quantity are essential for agricultural productivity increase. This study can therefore shed more light on this debate be a seminal contribution on this topic, which is increasingly getting policy traction in other areas of the world as well.

Response: We appreciate your time in carefully reviewing our study protocol and for recognizing the importance of this program evaluation, not only for India but around the world. In addressing your thoughtful comments, we feel the manuscript has been greatly improved. Thank you.

2. The agro-ecological interventions need to be spelt out better. There is a lot going on in terms of involving NGOs, encouraging women SHGs, and other community members to increase adoption. Each of these need to clearly explained. 

Response: We have included a Template for Intervention Description and Replication (TIDieR) checklist for population health and policy interventions in the supplementary materials.1 We have also added a figure that provides more detail about the program’s implementation. Of note, a process evaluation will be conducted to ensure we know exactly what activities are undertaken as part of the program in each BLOOM village and the actual exposure of farmers to these activities.2 A description of this process evaluation has also been added. (Supplementary Table S1, Fig 1, p. 29, lines 590-605)

3. While the agricultural impacts are easily understood, how (and why) the intervention is going to affect food consumption pattern is difficult to discern here. Is it through income or diversification of agriculture? I do not see a discussion on that.

Response: An agricultural intervention can affect dietary intake through several different pathways. The most important pathways are, as you note, (1) production, which increases access to food, and (2) income, which can increase expenditures on food. Other potential pathways that have been explored in the literature include: (3) women’s empowerment; (4) strengthening local institutions and thus service delivery; (5) improving availability and affordability of produce in local markets; and (6) improving knowledge of nutrition, health and WASH (specifically for nutrition-sensitive agricultural interventions). The evidence to support these pathways was systematically reviewed in 2020 (see Sharma et al. “Nutrition-Sensitive Agriculture: A Systematic Review of Impact Pathways to Nutrition Outcomes” Adv Nutr 2021;12:251-275).3 We have added this detail to the revised manuscript. (p. 5, lines 120-125)

4. Similarly, I am not able to understand the metrics chosen for health outcomes and the logic behind them.

Response: The health metrics chosen were based on a review of the evidence of the health effects of our two primary outcomes: (1) organophosphate insecticide exposure and (2) dietary diversity. We have added a table summarizing the evidence for each of the health outcomes assessed in the BLOOM study. (Table 1)

5. Also, is this a reasonable enough time to measure changes in health?

Response: The health outcome metrics, all continuous variables, were purposefully chosen because they are sensitive to short-term changes in exposure. We will also store biological specimens for intermediate disease biomarker analysis in future ancillary studies. Ultimately, BLOOM will provide robust baseline data with 2 years of follow up and the opportunity to follow these households over decades. Without this baseline, the chance to conduct a randomized controlled program evaluation would be lost as the government is quickly planning to scale up to all farmers across the state.

6. Some change in health status could also emerge from the changing labor requirements post the intervention. How is that accounted for?

Response: This is an excellent point. A previous study in Thailand found that musculoskeletal pain was worse in transitioning organic farmers than conventional farmers, potentially due to longer working hours and more manual labor tasks.4 However, another study in Finland did not find any difference in musculoskeletal pain between organic and conventional farmers.5 Anecdotally, in Andhra Pradesh where the BLOOM study is being conducted, organic farmers claim that weeding is actually easier because the soil is softer than under conventional farming systems. However, carrying natural farming inputs (heavy liquids carried on the head) may be more strenuous in comparison to conventional farming.

To ensure we capture this potential adverse effect of transitioning to organic farming, we will carefully measure both time use during the agricultural season and self-reported musculoskeletal pain in both men and women. We will also be collecting data on expenditures on agricultural labor and income from non-agricultural sources. The reason the latter is important is that increases in agricultural labor could adversely affect earning potential by reducing time available for other income-generating activities. Together, these data will enable us to evaluate how labor requirements change post-intervention and the impact this has on musculoskeletal pain and net household income, as well as differences in these impacts between men and women.

We have summarized the metrics, health outcomes, hypothesized pathways, and evidence in a new Table 1.

7. When NGOs are embedded in implementing an RCT, as it is proposed here, there is likely to be a higher take up of the programs creating a biased impact. I suspect, it is likely to be a case here. For reference, see Faraz Usmani, Marc Jeuland, Subhrendu K. Pattanayak; NGOs and the Effectiveness of Interventions. The Review of Economics and Statistics 2022.

Response: Thank you for sharing this fascinating article with us. We apologize for the confusion about who is implementing the intervention for the BLOOM study. This is not in fact an NGO. Rythu Sadhikara Samstha (RySS) is an agency established by the Department of Agriculture in Andhra Pradesh specifically to implement the Andhra Pradesh Community-managed Natural Farming (APCNF) program, which we are evaluating. It is therefore part of the state government and not an NGO. RySS is responsible for scaling up the APCNF program to all 6 million farmers across the state. A Memorandum of Understanding between the University of Edinburgh and RySS, signed in April 2021, has enabled a partnership in 2 districts (now 4 districts) wherein RySS has permitted us to randomize when 80 villages receive the training – starting in rabi 2023 (“intervention clusters”) or 24 months later (“control clusters”). Per our request, RySS gave us a list of villages in these districts where they had not conducted any prior trainings. We randomly selected 80 villages for inclusion in the BLOOM study from this list of “fresh villages.” Therefore, by design, the “NGO reputation effect” described in Usmani et al., which was seen in communities that had engagements with the NGOs prior to the intervention, would not apply in the case of the BLOOM study. In addition, a completely separate team from that implementing the intervention (RySS) is conducting the RCT (Public Health Foundation of India / University of Edinburgh).

The following text has been added to the revised manuscript:

A list of clusters in which APCNF has not yet been implemented will be provided by RySS and the evaluation team will randomly select clusters for inclusion in the BLOOM study from this list using simple random sampling. By purposefully enrolling clusters where RySS has never before worked, we will minimise what has been described elsewhere as the ‘[NGO] reputation effect’ [44]. In a previous randomised controlled trial in India, this reputation effect – meaning prior engagement with target communities – was shown to bias intervention effectiveness by at least 30% [44].

(p. 15, lines 286-289)

References

1. Campbell M, Katikireddi SV, Hoffmann T, Armstrong R, Waters E, Craig P. TIDieR-PHP: a reporting guideline for population health and policy interventions. BMJ 2018; 361: k1079.

2. Carroll C, Patterson M, Wood S, Booth A, Rick J, Balain S. A conceptual framework for implementation fidelity. Implement Sci 2007; 2: 40.

3. Sharma IK, Di Prima S, Essink D, Broerse JEW. Nutrition-Sensitive Agriculture: A Systematic Review of Impact Pathways to Nutrition Outcomes. Adv Nutr 2021; 12(1): 251-75.

4. Nankongnab N, Kongtip P, Tipayamongkholgul M, Bunngamchairat A, Sitthisak S, Woskie S. Difference in accidents, health symptoms, and ergonomic problems between conventional farmers using pesticides and organic farmers. Journal of agromedicine 2020; 25(2): 158-65.

5. Mattila TE, Rautiainen RH, Hirvonen M, Väre M, Perkiö-Mäkelä M. Determinants of good work ability among organic and conventional farmers in Finland. Journal of agricultural safety and health 2020; 26(2): 67-76.

---

## [Decision Letter · Decision Letter 1]

30 Jan 2023

Co-Benefits of Largescale Organic farming On huMan health (BLOOM): Protocol for a cluster-randomised controlled evaluation of the Andhra Pradesh Community-managed Natural Farming programme in India

PONE-D-22-22524R1

Dear Dr. Jaacks,

We’re pleased to inform you that your manuscript has been judged scientifically suitable for publication and will be formally accepted for publication once it meets all outstanding technical requirements.

Kind regards,

Arnab K. Basu

Academic Editor

PLOS ONE

Additional Editor Comments (optional):

Reviewers' comments:

Reviewer's Responses to Questions

**Comments to the Author**

1. Does the manuscript provide a valid rationale for the proposed study, with clearly identified and justified research questions?

Reviewer #1: Yes

2. Is the protocol technically sound and planned in a manner that will lead to a meaningful outcome and allow testing the stated hypotheses?

Reviewer #1: Yes

3. Is the methodology feasible and described in sufficient detail to allow the work to be replicable?

Reviewer #1: Yes

4. Have the authors described where all data underlying the findings will be made available when the study is complete?

Reviewer #1: Yes

5. Is the manuscript presented in an intelligible fashion and written in standard English?

Reviewer #1: Yes

6. Review Comments to the Author

You may also provide optional suggestions and comments to authors that they might find helpful in planning their study.

Reviewer #1: I am very satisfied with the

The responses provided by the author(s) are excellent which allay all of the concerns I raised in the first round of reviews.

7. PLOS authors have the option to publish the peer review history of their article (what does this mean?). If published, this will include your full peer review and any attached files.

Reviewer #1: **Yes: **Andaleeb

---

## [Editor Report · Acceptance letter]

17 Feb 2023

PONE-D-22-22524R1 

Co-Benefits of Largescale Organic farming On huMan health (BLOOM): Protocol for a cluster-randomised controlled evaluation of the Andhra Pradesh Community-managed Natural Farming programme in India 

Dear Dr. Jaacks:

I'm pleased to inform you that your manuscript has been deemed suitable for publication in PLOS ONE. Congratulations! Your manuscript is now with our production department. 

Kind regards, 

on behalf of

Dr. Arnab K. Basu 

Academic Editor

PLOS ONE